# Dental Fear and Anxiety of Kindergarten Children in Hong Kong: A Cross-Sectional Study

**DOI:** 10.3390/ijerph17082827

**Published:** 2020-04-20

**Authors:** Madeline Jun Yu Yon, Kitty Jieyi Chen, Sherry Shiqian Gao, Duangporn Duangthip, Edward Chin Man Lo, Chun Hung Chu

**Affiliations:** Faculty of Dentistry, The University of Hong Kong, Hong Kong, China; yonjunyu@hku.hk (M.J.Y.Y.); kjchen@hku.hk (K.J.C.); sherryg@hku.hk (S.S.G.); dduang@hku.hk (D.D.); hrdplcm@hku.hk (E.C.M.L.)

**Keywords:** dental fear, dental anxiety, children, behaviour, outreach

## Abstract

*Objectives:* The objectives of this cross-sectional study were to investigate the fear level of kindergarten children in the general population during dental outreach in a familiar kindergarten setting, and to explore the factors associated with the dental fear of kindergarten children. *Method:* Consecutive sampling method was used to select kindergarten children aged 3 to 5 to participate in a questionnaire survey and an outreach service. A behavioural observation type of instrument for dental fear and anxiety assessment—Frankl Behaviour Rating Scale (FBRS)—was chosen to investigate the fear level of the children. Bivariate analyses between various factors and children’s dental fear and anxiety were carried out using Chi-square test. *Results:* A total of 498 children participated in this study. Almost half (46%) of the children have had caries experience, and the mean dmft score was 2.1 ± 3.4. The prevalence of dental caries was 32%, 43%, and 64% in the 3-, 4- and 5-year-olds, respectively. Only 4% of the children scored negatively for dental fear and anxiety (95% CI 2.3%–5.7%). Children at three years of age displayed more dental fear and anxiety than children of older ages, but the difference in dental fear and anxiety among the genders and caries status was not statistically significant. Most of the children (92%) brushed daily, but only 20% of them used toothpaste. Most (85%) of them had never visited the dentist, and over 70% of them were mainly taken care by their parents. High levels of positive and cooperative behaviour and low levels of fear were found in this population. No statistical significance was found between the child’s dental fear and any factors except age. *Conclusion:* Children generally displayed low fear or anxiety levels in a dental outreach consisting of a non-invasive oral examination and preventive treatment in a familiar kindergarten setting. Conducting regular outreach dental services to kindergartens by providing oral examination and simple remineralisation therapies could be a promising strategy to not only control childhood caries, but also manage and reduce dental fear and encourage long term dental attendance in line with the medical model.

## 1. Introduction

Dental fear and anxiety are known deterrents to regular dental attendance [1,2]. Fear and anxiety constitute an important theme in dentistry, especially with children. In children, dental fear and anxiety affect their perception of dental care and alter their ‘first impression’ of receiving dental care. When children are surveyed regarding their dental fear and anxiety, those who fall in the category of ‘borderline’ might react inconsistently during potentially fear-provoking situations in the dental clinic and were regarded as at risk of developing long-lasting dental anxiety [3], which might have long-term consequences such as dental avoidance and even oral health neglect, as described in the vicious cycle of dental anxiety [4].

Dental fear and anxiety are major barriers to providing dental services to children and also encourages dental avoidance behaviour that could last till adulthood. In the literature, the proportion of the child population suffering from or at high risk of dental fear and anxiety ranged from around 6% to 20%, depending on the method of study, criteria of the study, and age of children in question [1,3,5,6,7,8,9]. For example, a study conducted in Japan classified around 8% of children aged 5 to 12 to be fearful based on their behaviour in a dental clinic [10], whereas 11% of 5-year-olds were found to be anxious in the United Kingdom when their parents were surveyed, and the proportion increased up to 21% for older children at 5 to 8 years of age in Taiwan by using a parental questionnaire [6].

In Hong Kong, a behavioural assessment for children’s dental fear has been previously attempted in a dental hospital setting with both patients from the general population or referrals from general practitioners. The prevalence of dental fear and anxiety in this sample population of 3- to 5-year-olds was found to be 8% [5], but non-dental visitors are automatically excluded and give little information about dental fear and anxiety in the general population of preschool children of the same age. Studies which were conducted on patients of paediatric specialist clinics have a much higher proportion of fearful children, presumably due to the highly selective population group in these studies [11,12,13], as well as studies which were conducted during fear-provoking clinical sessions rather than a simple check-up and non-invasive treatment [14,15,16]. It appears that the prevalence of dental fear and anxiety in children varies greatly between different countries and changes with the child’s age, as well as depending on the selected sample population.

In Hong Kong, kindergarten is pre-primary education and may be supported by government subsidy. It is provided to children from three years of age and lasts for three years, denoted as K1, K2, and K3 [17]. Although not compulsory, the enrolment rate of KG education was about 100%, i.e., virtually all children in Hong Kong attend kindergarten [18], after which primary school education from the age of six is mandatory. In terms of dental attendance, only a quarter of children had visited a dentist by the age of five, most of which were of symptomatic attendance [19]. Comparatively, caries’ prevalence in five-year-olds was as high as 55% during the same period [20]. Many children only start receiving dental care through a government-initiated, territory-wide dental care programme starting from Primary One level, which includes services such as check-ups and simple fillings [21]. At such a young age, strong emphasis should be placed on the prevention, control, and reversal of dental caries if life-long caries-free dentition is to be maintained, in accordance with the medical model of caries management [22]. 

In view of the high prevalence of early childhood caries and the need for early intervention in Hong Kong, the Faculty of Dentistry of the University of Hong Kong has been routinely organising outreach services to kindergartens, providing primary preventative care which includes oral health education, dental check-up and topical fluoride therapy to kindergarten children [23]. Silver diamine fluoride has been used in this case to arrest cavitated early childhood caries with parental consent. The use of silver diammine fluoride as the topical fluoride agent is quick to administer, and no caries excavation or surgical procedures are required [24], which is in turn less likely to provoke fear in children. It is known that dental fear and anxiety often arises and is reinforced by past negative dental experiences [11] and has been further shown to be associated with having dental diseases at a young age, such as caries [1]. However, sequential positive experiences or non-invasive treatment, which provokes little pain, such as that offered in kindergarten-based outreach dental services, might reduce dental fear and anxiety, explained by the latent inhibition theory [8,25]. The dental outreach primary preventive care can be a good first point of introduction to dentistry for young children in terms of both therapeutic and psychological advantage. 

To date, there are little data on the fear levels of young children in Hong Kong. Few studies on dental fear and anxiety of children have been conducted in outreach settings to date. We are interested in knowing the prevalence of dental fear and anxiety of children in Hong Kong at the level of preschool education. It was hypothesised that non-invasive dental check-up and caries treatment with the use of topical fluoride therapy in a familiar school environment is related to a low level and prevalence of dental fear and anxiety in preschool children. 

The objectives of the present study were to investigate the fear level of preschool children in the general population during dental outreach in a familiar kindergarten setting, and to explore the factors associated with the dental fear of preschool children.

## 2. Materials and Methods 

This cross-sectional observational study was conducted in Hong Kong from December 2019 to January 2020. Ethics approval was sought and obtained from the local Institutional Review Board (UW 19-611, HKU/HA HKW) before the commencement of the study. The reporting of this study follows the guidelines in the STROBE statement [26].

### 2.1. Sample Size

The sample size for the study was calculated based on an expected prevalence of 8% [10] in the population. The desired precision of estimation was set as 5%. With the confidence interval set as 95%, 452 children were needed in this study. In addition, estimating a 90% response rate, at least a total of 502 children needed to be invited to join this study.

### 2.2. Setting & Sample

Consecutive sampling method was used to select kindergartens from a list of all kindergartens in the said school year available from the website of Education Bureau. All students aged 3 to 5 studying from the K1 to K3 level of kindergarten were invited to participate in the study. An invitation letter and details regarding the content and procedure of the study were sent to the parents. Those who gave written consent for participation were included in the study. Inclusion criteria were generally healthy children aged 3 to 5. Exclusion criteria were children with severe systemic diseases, known behavioural or mental disorders, or who were absent on the day of the school outreach service. There were two parts to the study: kindergarten outreach and questionnaire survey. 

### 2.3. Fear Assessment and Clinical Exam

The assessment of dental fear and anxiety and dental clinical examination was conducted during the outreach service in the kindergarten setting. A behavioural observation type of instrument for dental fear and anxiety assessment—Frankl Behaviour Rating Scale (FBRS)—was chosen for this study [27,28]. The FBRS was used by the dental operator, who gave a score of 1 to 4 based on the behaviour of the subject (Table 1).

This scale was chosen as the behavioural observation method for dental fear and anxiety assessment was the most valid method with very young children, for which self-report questionnaires would be too complex to administer as a certain ability in the comprehension of the children is required. In contrast, parental proxy methods would likely be prone to bias from parents’ own perceptions because of the young age of the children [29]. Physiological monitoring would prove much too fear-provoking compared with the non-invasive check-up and preventive treatment that was to be delivered during the kindergarten outreach [30].

Before conducting the study, a single examiner (M.J.Y.Y.) was trained in the field and calibrated both clinical examination and fear assessment with an expert (D.D.) who was trained in paediatric dentistry and is a specialist in dental public health. On the day of the study, the examiner made the same short introduction to the waiting children. They then underwent dental clinical examination one by one by the same dentist (M.J.Y.Y.). Dental health status was recorded using decayed, missing and filled teeth (dmft) based on the presence of cavitation [31] and consequences of untreated caries (pulp involvement, ulceration, fistula, and abscess). Silver diammine fluoride was applied topically with a microbrush to carious lesions only. The dentist rated the child’s behaviour during the examination procedure using the FBRS. To improve the reliability of the outcome measures, the outreach team comprising the dentist and assistants was kept the same for all kindergarten visits in this study.

### 2.4. Questionnaire Survey

The questionnaire survey to parents of the kindergarten children was conducted about factors related to dental fear and anxiety of their children before the outreach service. The questionnaire explored: (a) the oral health-related behaviour of the child such as tooth brushing behaviour, use of toothpaste, snacking behaviour and dental attendance; and (b) dental experience of the child’s parent such as parent’s own dental fear, their dental visit experience, their satisfaction of child’s dental health and teeth appearance. Regarding the parents’ own dental fear, they were asked if they were afraid to see a dentist, with four possible answers on an ordinal scale: not afraid, slightly afraid, fairly afraid, and very/extremely afraid. Finally, demographic information such as the child’s sex and age, the child’s medical history, parents’ education level and their household income were collected.

### 2.5. Data Entry and Statistical Analysis

Data from the outreach and questionnaire survey were entered into a spreadsheet (Microsoft^®^ Excel für Mac, Version 16.35, Microsoft Corporation, Redmond, WA, USA) and cleaned before analysis. Dental fear and anxiety-related behaviours of the child subjects were described, and bivariate analysis between various factors and children’s dental fear and anxiety expressed in terms of their behaviour was performed using Chi square test. The programme Statistical Package for Social Science version 25.0 (IBM Corp., Armonk, NY, USA) was used for the statistical analysis.

## 3. Results

A total of 587 children attending K1 to K3, from five kindergartens, were invited to this study. Thirty-five children did not participate in this study, and written consent was obtained from 552 children. A total of 548 questionnaires were collected. Eighteen questionnaires had incomplete or invalid responses, and hence 530 valid questionnaires (with fewer than three invalid responses) were obtained. A total of 521 children attended the examination, and 33 children were absent on the day of dental outreach. A total of 498 children attended the oral examination with returned valid questionnaires. Hence, the participation rate of this study was 85% (498/587). A flow chart of the sample population is described in Figure 1 below.

The 498 children belonged evenly to the K1 (*n* = 174, 35%), K2 (*n* = 164, 33%) and K3 (*n* = 160, 32%) classes. Half (50%) of them were females. The great majority (95%) of the children were born in Hong Kong. Regarding their oral health-related behaviours, most (*n* = 458, 92%) of the children brushed at least once a day, but only 100 children (20%) used toothpaste. Most (*n* = 448, 90%) children snacked daily, and 75 (15%) children snacked three times or more per day. Most (*n* = 423, 85%) of them had never visited a dentist or had no dental experience. Most (*n* = 354, 70%) children were mainly taken care of by their parents.

From clinical oral examination, almost half (46%) of the children had caries experience and the mean DMFT score was 2.1 ± 3.4. The prevalence of dental caries was 32%, 43% and 64% in the 3-, 4- and 5-year-olds, respectively. Only 4% of the children scored negatively for dental fear and anxiety (95% CI 2.3%–5.7%). Children at three years of age displayed more dental fear and anxiety than children of older ages, but the difference in dental fear and anxiety among the genders was not statistically significant. Further Bonferroni correction revealed that 3-year olds had significantly more negative or definitely negative behaviour scoring than 4-year-olds (*p* = 0.002) and 5-year-olds (*p* < 0.001), but no statistical significance was found between the 4- and 5-year-olds. The distribution of FBRS scores among different ages and gender is shown in Table 2.

A child’s behaviour was categorised as ‘negative’ (FBRS score 1 and 2), and ‘positive’ (FBRS score 3 and 4) before bivariate analysis was carried out, with factors related to child oral health-related behaviours or hospitalisation experience, as well as parental ratings of child’s oral health, dental appearance, and demographic data. Parents’ own dental experience and fear were analysed with their children’s FBRS scores based on who answered the questionnaires by Chi square tests; no statistical significance was found (Table 3). Multivariate analysis using logistic regression was not performed in the end as statistical significance was found in only one factor (age) with FBRS score, and there were no cases with negative scores for 5-year-olds.

## 4. Discussion

Dental fear and anxiety levels was found to be low, at 4%, among kindergarten children in Hong Kong based on behavioural observation during school-based outreach. Although this was the first time that most children had seen a dentist or received a dental check-up during kindergarten, children displayed few fearful behaviours during the outreach. This is possibly greatly due to the nature of the outreach arrangement—it was conducted in a familiar kindergarten setting on an ordinary school day in the presence of teachers and classmates whom children knew well—as well as the non-invasiveness of dental check-up and topical fluoride therapy. Each child participant took an average of one minute to go through the entire check-up process. These factors might have kept the fear and anxiety of children at low levels. 

Age was found to be an influential factor on a child’s expressed dental fear during the outreach, whereas a gender difference was not apparent. Younger preschool children, particularly those in their first year of school in K1, tended to express more dental fear and anxiety, congruent with what was found in previous studies [11,32,33]. It is likely that additional years of schooling helped children to follow instructions and engage in a new experience of dental check-up and preventive fluoride treatment, as well as improving their communication ability with the dentist and assistants. It is also possible that older children have had previous dental experience and were familiar with dental personnel before the school-based outreach.

Other investigated factors, including dmft as a dental health status, did not show a statistically significant association with dental fear and anxiety-related behaviour. This is in part due to the small proportion of individuals classified as ‘fearful’ based on their behaviour in the non-invasive, school-based dental visit. Although sufficient as an epidemiological investigation on the level of dental fear and anxiety in preschool children in Hong Kong, the low prevalence of dental fear and anxiety meant that further studies involving more fearful children would be necessary for valid comparisons between groups of fearless and anxious children. Such a cross-sectional study also serves as a snapshot only, but a causal relationship between dental outreach and dental fear and anxiety could be ascertained with additional follow-up.

An appropriate choice of dental fear and anxiety assessment in children was essential in the study and had a strong influence on the interpretation of study results. The chosen scale, FBRS, is a behavioural scale used for measuring dental fear and anxiety, which is different from other forms of assessment, such as self-report, parental proxy and physiological measurement [34]. The method of carrying out an assessing using FBRS was able to capture the dental fear and anxiety of participating children at the scene, in other words, the state anxiety. Compared with other measures of dental fear and anxiety, the FBRS does not require extra feedback from the children, and avoids the requirement of children having the cognitive ability to answer scenario-type questions themselves [5,35]. The FBRS also does not require monitoring equipment to be connected to participating subjects, which could be fear-provoking themselves [30]. Furthermore, as a commonly used behavioural observation method, a high correlation and significant relationship have been found between FBRS and various dental fear and anxiety assessment measures [36,37,38]. This being said, it is beneficial to note that a study of a similar nature conducted in a different setting and on children of an older age group would remove the relevant restrictions and allow the use of a variety of other common assessment tools as well, such as the CFSS-DS, as a form of self-report, and a psychometric scale to complement the shortcomings of using the FBRS alone.

Understanding the limitations of the current study aids in the interpretation of results and planning of subsequent follow-up studies. In the investigation, rating scores could not be duplicated because of the nature of the study. Different rating scores were expected as fear and anxiety levels change and children behave differently upon their first and subsequent experience in dental check-up. Videotaping the process for the purpose of assessing intra-rater reliability was not feasible in the kindergarten setting. The same trained examiner has been carrying out the data collection in order to increase the reliability of the data collected.

The present study covered children ranging from K1 to K3 of kindergarten in Hong Kong. The proportion of dental fear and anxiety that resulted from the study serve as a useful reference for the pre-primary population in Hong Kong. The low dental fear and anxiety levels exhibited by young children at dental outreach programmes demonstrate that such outreaches are not only useful in dental health surveillance and disease prevention aspects, but also possibly effective in maintaining and reinforcing positive dental experiences for children. Further tracing of children exposed to regular dental check-up and/or preventive dental treatment arranged on school campus in a prospective study design may unveil how kindergarten-based dental programmes modulate and possibly reduce the dental fear and anxiety of young children.

## 5. Conclusions

Children generally displayed low fear or anxiety levels in a dental outreach consisting of non-invasive oral examination and preventive treatment in a familiar kindergarten setting. The study revealed that younger kindergarten children exhibit higher levels of dental fear and anxiety than older children. Dental outreach in kindergarten settings is recommended as a child’s first experience of dental check-up and preventive care, for reasons of dental health maintenance and building of positive attitudes towards dentistry. Future studies of prospective nature and investigations targeting children with known higher levels of dental fear and anxiety can be conducted to follow up children exposed to regular kindergarten-based dental outreach and enrich our understanding of factors related to the dental fear and anxiety of young children.

## Figures and Tables

**Figure 1 ijerph-17-02827-f001:**
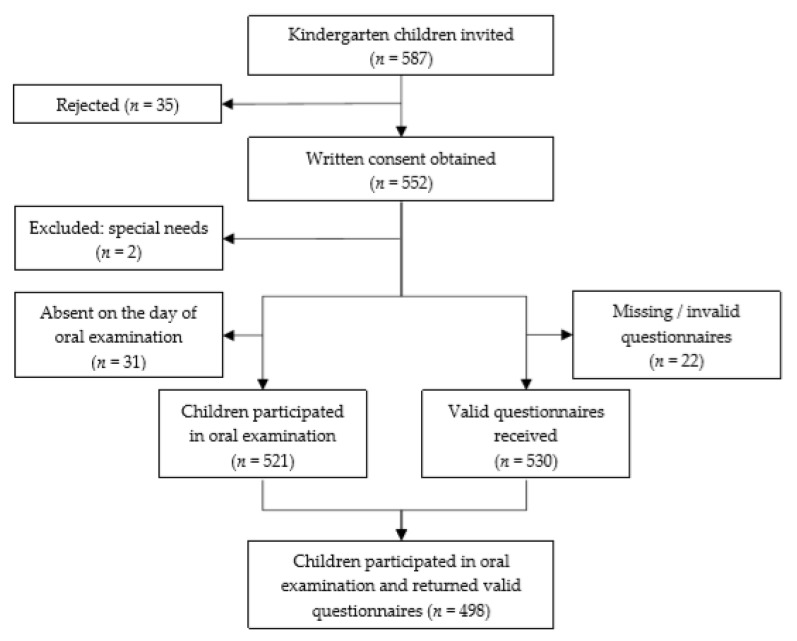
Children recruitment and participation in the study.

**Table 1 ijerph-17-02827-t001:** Frankl Behaviour Rating Scale (FBRS).

1	Definitely negative	Refusal of treatment, forceful crying, fearfulness, or any other overt evidence of extreme negativism.
2	Negative	Reluctance to accept treatment, uncooperative, some evidence of negative attitude but not pronounced (sullen, withdrawn).
3	Positive	Acceptance of treatment; cautious behaviour at times; willingness to comply with dentist, at times with reservation, but child follows dentist’s directions cooperatively.
4	Definitely positive	Good rapport with the dentist, interest in the dental procedures, laughter and enjoyment.

**Table 2 ijerph-17-02827-t002:** Frankl Behaviour Rating Scale (FBRS) score of children according to age and sex.

FBRS Score	1 Definitely Negative	2 Negative	3 Positive	4 Definitely Positive	*p*-Value
Age					<0.001
3	5 (2.9%)	12 (7%)	18 (10.5%)	137 (79.7%)	
4	2 (1.2%)	1 (0.6%)	23 (13.9%)	140 (84.3%)	
5	0 (0%)	0 (0%)	12 (7.5%)	148 (92.5%)	
Sex					0.822
Male	4 (1.6%)	8 (3.2%)	27 (10.8%)	210 (84.3%)	
Female	3 (1.2%)	5 (2.0%)	26 (10.4%)	215 (86.3%)	
Total	7 (1.4%)	13 (2.6%)	53 (10.6%)	425 (85.3%)	

**Table 3 ijerph-17-02827-t003:** Bivariate analysis between child’s behaviour and potential factors.

Factors	Negative Behaviour	Positive Behaviour	*p* Value
Caries experience (dmft > 0)			0.914
Yes	9 (45%)	221 (46%)	
No	11 (55%)	257 (54%)	
Consequences of untreated caries			1.000
Yes	3 (15%)	70 (15%)	
No	17 (85%)	408 (85%)	
Daily toothbrushing			1.000
Yes	19 (95%)	441 (92%)	
No	1 (5%)	37 (8%)	
Use of toothpaste †			0.247
Yes	6 (30%)	91 (19%)	
No	14 (70%)	387 (81%)	
Daily snack habit			1.000
Yes	18 (90%)	418 (87%)	
No	2 (10%)	60 (13%)	
Parents as main caregiver			0.768
Yes	15 (75%)	339 (71%)	
No	5 (25%)	137 (29%)	
Child’s dental experience †			1.000
Yes	3 (15%)	70 (15%)	
No	17 (85%)	408 (85%)	
Hospitalisation experience			0.402
Yes	10 (50%)	194 (41%)	
No	10 (50%)	284 (59%)	
Born in Hong Kong			0.711
Yes	19 (95%)	426 (81%)	
No	1 (5%)	52 (11%)	
Parent’s satisfaction of child’s dental health			0.121
Yes	5 (25%)	203 (42%)	
No	15 (75%)	275 (58%)	
Parent’s satisfaction of child’s teeth appearance			0.860
Yes	15 (75%)	350 (73%)	
No	5 (25%)	128 (27%)	
Father attained more than mandatory education			0.718
Yes	14 (70%)	354 (74%)	
No	6 (30%)	126 (26%)	
Mother attained more than mandatory education			0.347
Yes	12 (60%)	334 (70%)	
No	8 (40%)	144 (30%)	
Household income above median			0.798
Yes	15 (75%)	346 (72%)	
No	5 (25%)	132 (28%)	
Mother’s dental visit experience †			0.766
Yes	9 (69%)	288 (72%)	
No	4 (31%)	114 (28%)	
Mother’s dental treatment experience †			0.777
Yes	9 (69%)	256 (64%)	
No	4 (31%)	146 (36%)	
Mother’s own dental fear †			0.632
Afraid	2 (22%)	44 (15%)	
Not afraid	7 (78%)	247 (85%)	
Mother rescheduled or cancelled appointments before †			1.000
Yes	0 (0%)	25 (6%)	
No	13 (100%)	377 (94%)	

† Fisher’s exact test was used.

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
