# Peer review of "Dental Fear and Anxiety of Kindergarten Children in Hong Kong: A Cross-Sectional Study"

_ijerph, 2020, doi:10.3390/ijerph17082827_

Round 1
Reviewer 1 Report
This is an interesting paper investigating the prevalence of dental anxiety/fear in young children in a kindergarten setting. There are a few parts that need further improvement and clarifications.
[1] Page 1, line 20: "Only 4% of the children scored negatively for dental fear..." This sentence reads confusingly. It means children scored negatively in their behaviours instead of 'negatively for dental fear'. Please change the similar sentences in the Results section.
[2] Page 2, line 79-87: This paper has very little to do with SDF. However this part was presented with too much details about SDF, which is a bit deviating from the focus of examining the dental anxiety/fear of pre-schoolers.
[3] Page 3, section '2.3 Fear assessment and clinical examination': It is not clear whether the single dentist/investigator has been trained and calibrated. Also the paper does not address the intra-rater reliability test.
[4] Page 4, section '2.4 questionnaire survey': It is also not clear how mothers were assessed on their dental anxiety level. What scale has been used for assessing adult dental anxiety? The relevant details should be given in this section.
[5] Page 7, section 'Discussion': The paper should also specify the limitations of the current study.
Author Response
Thank you for your valuable suggestion. The manuscript has been revised accordingly. Please find the point-by-by response in the attached file.

Reviewer 2 Report
I am afraid there is a misunderstanding form the authors as the way dental fear can be measured.
The Frankl scale is an instrument to measure behaviour and not dental fear.
There are other instruments validated n several languages like the DFSS.
The paper needs rewriting.
Author Response
Thank you for your valuable comments. The manuscript has been revised accordingly. Please find a point-by-point response in the attached filed.

Reviewer 3 Report
Very interesting and novelty paper Dental Fear and Anxiety of Kindergarten Children.
The manuscript is well written. The introduction was sufficient. All materials and methods used were well defined. Images, tables, graphs, and other results were clearly presented. Conclusions drawn from the results were logical. I recommend minor revision:
Only: Abstract:
M&M: add statistical analysis
Results: “and the mean dmft score”. Add the significance of dmft
Author Response
Thank you for your suggestion. Our manuscript has been revised accordingly.

Round 2
Reviewer 2 Report
.....Furthermore, as a commonly used behavioural observation method, high correlation and significant relationship have been found between FBRS and various dental fear and anxiety assessment measures [35-37].
I still believe this not communicated correctly, even in the results section.
There is no direct correlation used for behaviour to extrapolate dental fear.
There is f.e. arbitrarily use of “definitely negative behaviour” as “dental fear”
The article you are referring(Children's fear and behavior in private pediatric dentistry practices. Baier et al . Pediatr Dent. 2004 Jul-Aug;26(4):316-21.) makes exactly this distinction.
Conclusion of this article :
The proportion of children with dental fear in private pediatric dentistry practices was 20%, and the proportion of children with negative behavior during treatment was 21%. Children with negative behavior had greater odds of having dental fear and children with dental fear had greater odds of having negative behavior.Screening for dental fear may allow pediatric dentists to prepare children more adequately for positive treatment experiences.
Author Response
Thank you for your valuable comments. We have incorporated your comments in our discussion. We also agree that using behavioural observation method has its limits and this has been acknowledged in the discussion and interpretation of results.
